# Morphology of ejecta features from the impact on asteroid Dimorphos

Fabio Ferrari [1] ✉, Paolo Panicucci [1], Gianmario Merisio [1],
Carmine Giordano [1], Mattia Pugliatti [1], Jian-Yang Li[2],
Eugene G. Fahnestock [3], Sabina D. Raducan [4], Martin Jutzi[4],
Stefania Soldini [5], Masatoshi Hirabayashi [6,7], Colby C. Merrill [8],
Patrick Michel [9,10], Fernando Moreno[11], Gonzalo Tancredi [12],
Jessica M. Sunshine [13,14], Jens Ormö [15], Isabel Herreros [15],
Harrison Agrusa [9,13], Ozgur Karatekin [16], Yun Zhang [17], Nancy L. Chabot [18],
Andrew F. Cheng [18], Derek C. Richardson [13], Andrew S. Rivkin [18],
Adriano Campo Bagatin [19], Tony L. Farnham [13], Stavro Ivanovski[20],
Alice Lucchetti [21], Maurizio Pajola [21], Alessandro Rossi [22],
Daniel J. Scheeres [23] & Filippo Tusberti [21]

Hypervelocity impacts play a significant role in the evolution of asteroids, causing material to be ejected and partially reaccreted. However, the dynamics and evolution of ejected material in a binary asteroid system have never been observed directly. Observations of Double Asteroid Redirection Test (DART) impact on asteroid Dimorphos have revealed features on a scale of thousands of kilometers, including curved ejecta streams and a tail bifurcation originating from the Didymos system. Here we show that these features result naturally from the dynamical interaction of the ejecta with the binary system and solar radiation pressure. These mechanisms may be used to constrain the orbit of a secondary body, or to investigate the binary nature of an asteroid. Also, they may reveal breakup or fission events in active asteroids, and help determine the asteroid's properties following an impact event. In the case of DART, our findings suggest that Dimorphos is a very weak, rubble-pile asteroid, with an ejecta mass estimated to be in the range of $(1.1-5.5) \times 10^7$ kg.

On September 26, 2022, NASA's Double Asteroid Redirection Test (DART) spacecraft impacted successfully Dimorphos, the smaller asteroid of binary system (65803) Didymos[1,2]. Ground and space-based observations of the impact event permitted determining the net effect of the impact on the orbital motion of Dimorphos[3–8], and identifying features on a scale of thousands of kilometers in the ejecta, such as an anti-solar tail and curved streams originating from the Didymos system[9–14]. Additional observations estimated the mass of the observable ejecta in the range $(1.3-2.2) \times 10^7$ kg (ref. 15), and $(0.9-5.2) \times 10^7$ kg (ref. 16), and identified a population of meter-sized boulders that escaped from the system with overall mass of $5.2 \times 10^6$ kg (ref. 17).

Estimates of ejecta mass from recent simulation works are in line with observations, and provide values of e.g., $9.4 \times 10^6$ kg by using remote observation data[18], or $(1.7-4.3) \times 10^7$ kg by matching the observed orbital change of Dimorphos[3] with the impact's momentum enhancement factor[4,19], or $6.5 \times 10^6 - 4.1 \times 10^7$ kg (ref. 20), using close-up images of the system taken by LICIACube[21–23]. The mass estimates above are likely lower-bounds, as they refer to sub-set of observable and/or simulated ejecta.

Here we discuss the dynamical origin of the long-range features observed, and estimate the mass, velocity and size range of ejecta particles involved in their formation. We analyze observations of the

DART impact by the Hubble Space Telescope (HST)[9] up to three weeks after the impact, and we study the dynamics of ejecta using numerical simulations and synthetic image generation to reproduce the observed features. We show that these features result naturally from the dynamical interaction of the ejecta with the binary system and solar radiation pressure (SRP).

## Results

Simulations were performed to reproduce the dynamics of 5 million ejecta particles in the $10^{-6}$–$10^{-1}$ m radius range, under a wide range of parameters and initial conditions (see "Methods"). We simulate the data acquisition from the HST by generating synthetic images that match acquisition time, geometry, and illumination conditions, and by reproducing the photometric output of HST observations for direct comparison (see "Methods"). Best-fit simulations are chosen by evaluating quantitatively the similarity between HST and synthetic images. This process is based on semi-automatic identification of features and a scoring mechanism applied to both HST and synthetic images (see "Methods").

Ejecta are initially arranged on the surface of a cone whose apex is located at the DART impact location on Dimorphos's surface[1], and centreline is directed opposite to the impact velocity vector of the DART spacecraft. We assume that the ejecta cone axis and momentum vector are aligned with the DART incident direction. Note that recent work on ejecta cone geometry showed this may not be the case, exhibiting complex geometry[24,25] as well as some clusters not following the cone[4,9,17]. However, such conditions appear to not impact the global evolution of ejecta and their long-range features significantly. For each simulation, we define a Velocity-Size Distribution (VSD, see "Methods" Eq. 3) to initialize the velocity of ejecta particles according to their radius. We assign particles in the cone a velocity between the escape speed from the Didymos system, and a maximum speed, which we set between 0.50 and 500 m/s (ref. 21), depending on the simulation set (see Supplementary Fig. 1).

We restrict our analysis to particles that do not re-accrete immediately to Dimorphos after impact, i.e., particles that are initially faster than the escape speed from Dimorphos (8.4 cm/s, refs. 19,26,27). The evolution of ejected material is driven mainly by gravitational interactions with Didymos and Dimorphos, by impacts and re-accretion on the asteroids' surfaces, and by the effect of Sun's Radiation Pressure (SRP) and tides[9,28–30]. The dynamics of particles are dominated by gravity near the asteroid system, and by SRP farther away. Because SRP is less effective for larger particles, this switch occurs at ~10 km from Didymos system for cm-sized particles, and at about 2 km for mm-sized ones[28]. Particles smaller than $10^{-4}$ m in size are dominated by SRP immediately after they are ejected from Dimorphos and do not interact significantly with Didymos gravity field[28]. This was also observed by HST, which showed the formation of a SRP driven tail a few hours after the impact[9].

Our simulations show that all ejecta features observed can be related to two main evolutionary paths: the evolution of the ejecta cone, and the formation of the tail. Individual numerical values reported in the followings refer to our best-fit solution (see "Methods"). This shows that ~$1.7 \times 10^7$ kg of ejecta are involved in the formation and feeding process of the tail, with initial speeds up to a few m/s. The curved ejecta streams, i.e., the linear features gradually curving with time over the first 4 days after the impact[9], involve ~$2.4 \times 10^6$ kg of ejecta, with initial speeds up a few m/s. We find these streams are the end arms of a spiral motion originating in the inner system, as discussed below. Table 1 reports properties of ejecta in each feature, at time T0 + 11.86 days, with reference to regions labeled in Fig. 1. At that time, most of ejecta mass is contained within the inner system, with all particles with radius below 2 mm having already escaped the system. When not impacting on either asteroid, these ejecta fragments act as a reservoir to feed the tail, as they are pushed towards the anti-solar direction progressively by SRP.

### Cone-related features

The dynamical evolution of the ejecta cone is perhaps the most unique aspect of DART's impact on Dimorphos. The binary nature of Didymos makes the evolution of the ejecta cone very different from that observed in other natural impact-driven events, e.g., asteroid Scheila[31,32] or comet P/2010 A2[33–36]. DART's ejecta are produced ~1.2 km away from the barycenter of Didymos system, and carry a translational velocity of 0.169 m/s due to the orbital motion of Dimorphos around the primary[37,38]. This makes slow ejecta with speed lower than 2 m/s to interact gravitationally with Didymos, forming an outwards clockwise spiraling structure about the barycenter of the system[28] (see Supplementary Video 1), which was also observed by HST[9]. Observations of the spiral's evolution constrain the mass of ejecta involved, as the spiral appears to be visible in synthetic images only above the critical mass of ~$10^6$ kg. In addition, the timing of the spiral motion constrains dynamical quantities of ejecta, such as their initial velocity and size range. Ejecta slower than Didymos escape speed (24.8 cm/s from Dimorphos's impact location[26,27]) are launched into elliptic orbits about Didymos. Faster ejecta escape the system on hyperbolic arcs and are visible in the early HST images (Fig. 2). The orbit of ejecta is perturbed due to close interactions or collisions with Didymos or Dimorphos, SRP, and solar tides. We find the spiraling motion observed by HST involves ejecta particles larger than 295 μm, with a differential size-frequency distribution (dSFD, see "Methods" Eq. 4) of index −2.28, and velocities up to 2.00 m/s. Both the radiometric (to constrain mass and size distribution) and dynamic (to constrain velocity) analyses (see "Methods"), suggest that the properties of ejecta in the southern arm of the spiral differ slightly from those in the northern arm. We find that the southern arm is about 20% more massive than the northern arm and contains slower and larger particles, as reported in Table 1. This effect is due to the Sun-relative geometry, which heavily affects the dynamics of smaller ejecta fragments. In fact, the southern arm is directed nearly towards the Sun, while northern arm is approximately orthogonal to it. This makes the southern arm less developed in terms of area covered in the images, but denser, as the fragments are slowed down up to a point where their velocity is reversed in the anti-solar direction and redirected towards the Didymos system. The net effect is that less mass escapes the binary system from the southern direction. Conversely, the SRP acts orthogonally to the northern arm, with the effect of spreading the ejecta over a larger area, decreasing their surface density[9]. In addition, ejecta particles in the northern arm fly by closer to Didymos and more particles interact gravitationally with it compared to those in the southern arm[21,24,25]. This increases further the spreading effect of ejecta onto a larger area and multiple directions around the system.

Figure 2 shows a direct comparison between synthetic images from numerical simulations and HST observations. Simulations reproduce the general morphology and dynamical evolution of features observed by HST (apart from inhomogeneous individual features[39,40], see Supplementary Fig. 2) and allow us to identify their dynamical origin. Figure 2a–d shows two curved streams (s1, s2) at T0 + 1.14 days originating from the southern part of the image and then spiraling out to feed the northern arm of the spiral. The spiral arcs overlap with the faster material in the outgoing ejecta cone (c1-2 southern; c3-4 northern) and in the tail (t1-2), while the southern edge of the tail (t3) origins from the spiral motion itself. Figure 2e–h shows the linear features (l1, l2) related to the clearance of ejecta after the spiral has evolved, and the edges of the tail (t1, t3), with its bifurcating line (t2).

The dynamics of ejecta particles in the inner system reveal the feeding mechanism of the spiral and the tail. The spiral originates from the translational off-set produced by the initial location and velocity of Dimorphos along its orbit. Figure 3 shows the trajectory of particles that have not been ejected from the system after 11.86 days (larger than 4 mm), i.e., they are still within the Hill's sphere of Didymos

**Table 1 | Properties of ejecta features at T0 + 11.86 days (08-Oct-2022 19:57:00) for our best-fit solution (see "Methods")**

| Feature | Mass [kg] | Initial velocity [min, max] m/s | Radius range [min, slope] | Region of Fig. 1b |
|---|---|---|---|---|
| Spiral—northern arm | $2.4 \times 10^5$ | [0.22, 2.00] | [295 μm, −1.56] | Blue |
| Spiral—southern arm | $2.9 \times 10^5$ | [0.20, 1.76] | [376 μm, −3.16] | Dark green |
| Spiral—total | $5.2 \times 10^5$ | [0.20, 2.00] | [295 μm, −2.28] | |
| Tail | $1.1 \times 10^5$ | [0.20, 2.55] | [189 μm, −2.13] | Light green |
| Didymos system | $8.9 \times 10^6$ | [0.18, 0.28] | [2.23 mm, −1.77] | Yellow |
| Impacted on Didymos | $3.3 \times 10^6$ | [0.18, 0.90] | [1.90 μm, −2.09] | - |
| Impacted on Dimorphos | $4.4 \times 10^6$ | [0.19, 0.32] | [5.06 μm, −1.99] | - |
| Impacted—total | $7.7 \times 10^6$ | [0.18, 0.90] | [1.90 μm, −2.07] | - |
| Background | $1.9 \times 10^6$ | [0.19, 3.07] | [143 μm, −2.33] | Purple |
| Outside the FOV | $6.6 \times 10^3$ | [0.20, 19.22] | [1.00 μm, −2.41] | - |
| **Total** | $\mathbf{1.9 \times 10^7}$ | **[0.18, 19.22]** | **[1.00 μm, −2.70]** | |

We account for the field of view, geometry, illumination conditions and photometric performance of HST images. We report here values of mass, initial velocity range, and radius distribution associated to ejecta population labeled according to Fig. 1 (the last column reports the color of the corresponding region for reference): Spiral (particles in the features originated by the curved streams, i.e., northern and southern edges of the cone), Tail, Didymos system (particles in the central area of the image, this does not account for particles impacted on either asteroid), Impacted, Background (particles not linked to any feature), Outside the FOV (not in the image, already escaped out of the FOV). Bold lines between rows in are used to group features by families.

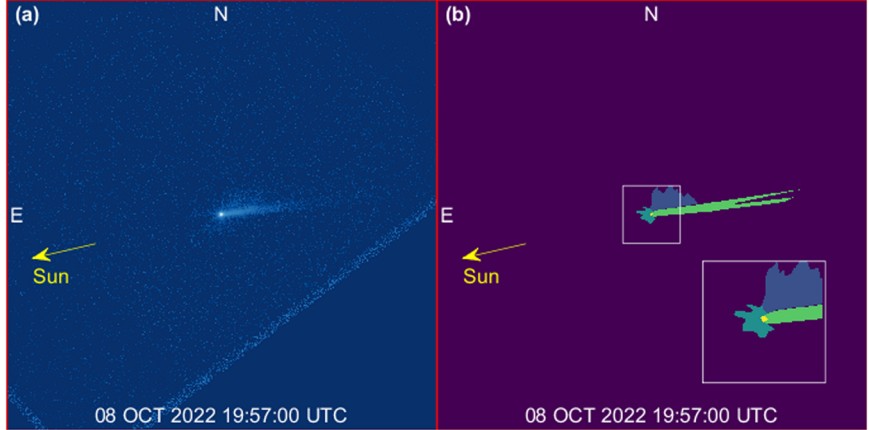

**Fig. 1 | Identification of features. a** HST image at T0 + 11.86 days (08 OCT 2022 19:57:00 UTC)[5]. Didymos system and ejecta features are visible in the central part of the figure. The oblique lines (bottom-right/-left part of the image) are artifacts. **b** Identification of the main ejecta features: northern (blue) and southern (dark green) arms of the spiral, tail (light green) and Didymos system (yellow). For better visualization of colored regions, a zoomed box is provided to the bottom-right part of the figure. Each frame is 6480 km wide, the inset at the bottom-right of (**b**) is 864 km wide.

(68 km)[26]. Three main outgoing directions are identified, aligned with the southern arm, northern arm, and tail, consistently with larger-scale images (Fig. 2). Southern and northern arms outlets contain particles larger than a few mm, while particles escaping through the tail direction are sensibly smaller. The elliptic orbits of cm-sized particles appear to be clustered in argument of pericenter along two directions, which are initially ~50 deg apart. One of the clusters rotates in the out of plane direction, and evolves towards a nearly Sun-Synchronous Terminator Orbit[41], while the other one rotates in the view plane to align with the sunward direction. Both behaviors are consistent with the perturbing action of SRP[28,29]. We observe that cm-radius particles ejected in the sunward direction return to the HST field of view 8–12 days after the impact.

## Tail-related features

As done for the cone-related features, we use a photometric analysis to constrain the mass of ejecta in the tail, and a dynamic analysis, based on timing and evolution of the tail, to constrain the size range of the ejecta.

We remark that the mass ejected and fed to the tail increases with time, as more ejecta particles escape the inner Didymos system. The timing of the tail formation constrains the size of particles inside the

tail, as their dynamics are driven by their ballistic coefficient $\beta_{SRP}$ only, which depends solely on their size (see "Methods"). The smaller fragments are ejected rapidly from the system due to SRP to form the early tail, while larger ones are bounded within the system for a longer time[26,42].

A secondary tail was observed between 5.7 and 18.5 days after the impact[9], rather appearing right north to the first tail within a position angle of 4 deg, while the first tail gradually rotates. A few possible explanations have been proposed to support the formation of the tail, including a secondary impact onto Didymos or Dimorphos, or abrupt disruption of a large boulder ejected after the impact[9,10,18]. Our simulations suggest that the gravitational interactions between slow ejecta and the Didymos binary system produce naturally bifurcations in the main tail, without the need of additional impacts or disruption events. This is also consistent with considerations on observations' viewpoint geometry[42]. While this finding does not reject other hypotheses, it describes a possible natural formation mechanism for secondary tails in a binary system. Close flybys of slow ejecta around Didymos produce individual streams of particles which are eventually injected in the direction of the tail, each at different time and from different directions. In our simulations, a sunward arm of the spiral appears at the beginning of the tail, leading to the formation of a secondary tail,

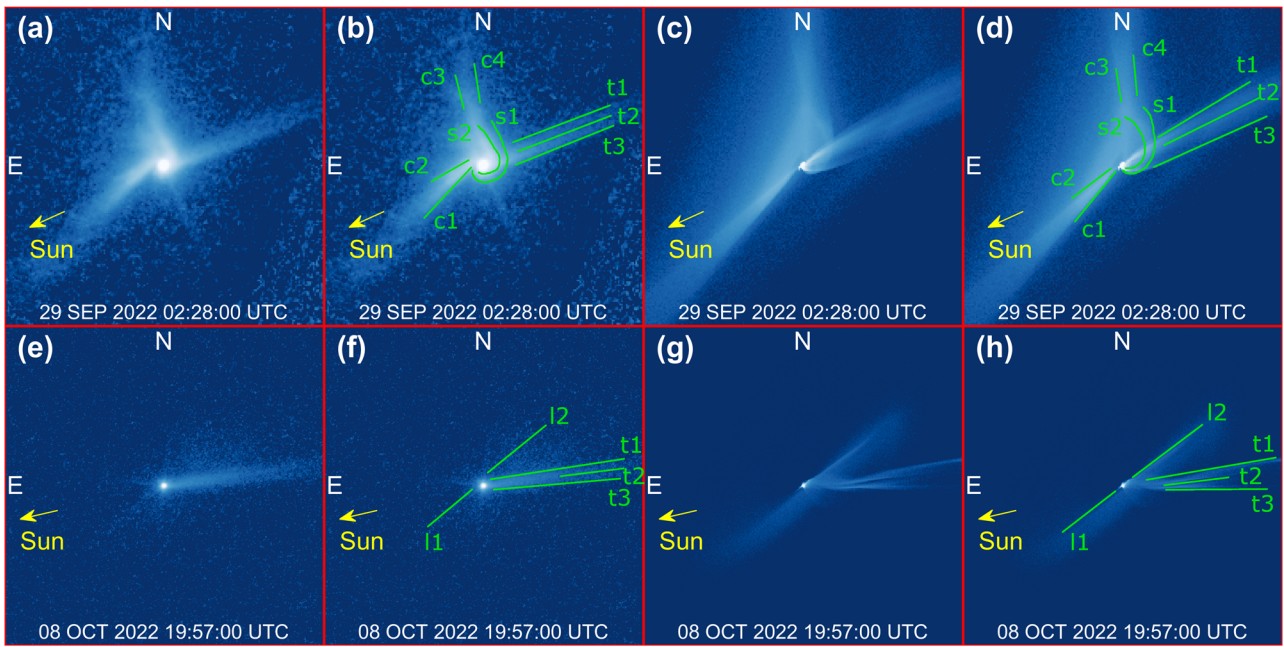

**Fig. 2 | Time evolution of features. a–d** Spiral feature at T0 + 1.14 days (29 SEP 2022 02:28:00 UTC) and (**e–h**) tail at T0 + 11.86 days (08 OCT 2022 19:57:00 UTC). The panel reports duplicated pairs of images (left unannotated, right annotated): (**a, b, e, f**) HST images[5], (**c, d, g, h**) synthetic images from numerical simulation (ID 02, see Supplementary Table 1). Annotations report curved streams (s1-2), cone edges (c1-4), tail edges and bifurcation line (t1-3), linear features (l1-2). Each frame is 864 km wide. See Supplementary Video 1 for full video sequence.

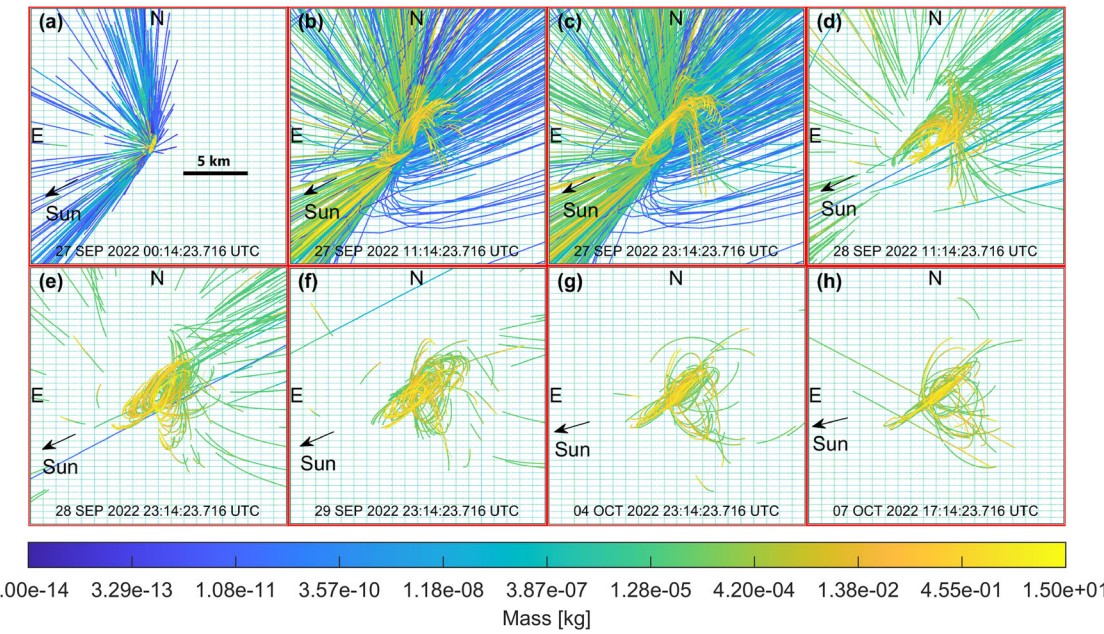

**Fig. 3 | Close up view of the inner system (20 km field of view). a–d** Spiral motion in the inner system and tail feeding mechanism involving particles that are still near the Didymos system after 14 days. Simulated trajectories are shown, with no photometric adjustments applied. **e–h** Argument of perihelion of orbits of ejecta with radius in the range 1–10 cm is clustered along two directions ~50 deg apart. Ejecta of cm radius returning to the system (HST view) between 8 and 12 days after the impact. See Supplementary Video 3 for full video sequence and Supplementary Videos 4–6 for larger scale views (FOV of 100 km, 1000 km, 5000 km). Figures refer to simulation ID 02 (see Supplementary Table 1).

which is clearly visible after 2.13 days (Fig. 4). This mechanism produces a small curvature at the beginning of the tail, which was also detected by HST[9]. We find this is a footprint of Didymos binary motion in the tail, as the curvature produces an oscillatory motion that propagates in the anti-Solar direction (see panel (h) of Fig. 5, where this effect is exaggerated for clarity, or Supplementary Video 2).

Oscillations have a period commensurate to the binary period, as this is the combination of the translational/orbital motion imparted by Dimorphos around Didymos, and SRP. This feature involves the least massive particles in the spiral: unlike most of ejecta in the cone that are ejected towards the southern or northern arms of the spiral, ejecta particles in the curved tail are ejected directly in the anti-Solar

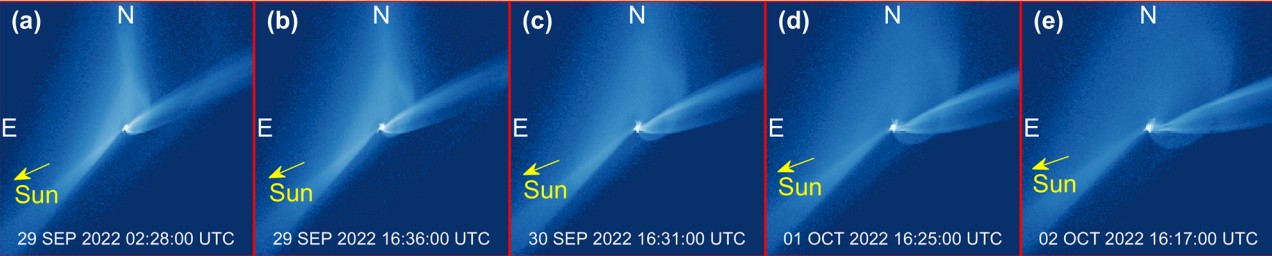

**Fig. 4 | Synthetic images showing the time evolution of spiral and tail features.** **a** Natural bifurcations in the tail and tail curvature appear evident starting from T0 + 2.13 days (29 SEP 2022 02:28:00 UTC). **b**–**e** Evolution of the bifurcation in the subsequent 4 days. Each frame is 864 km wide. Figures refer to simulation set ID 02 (see Supplementary Table 1).

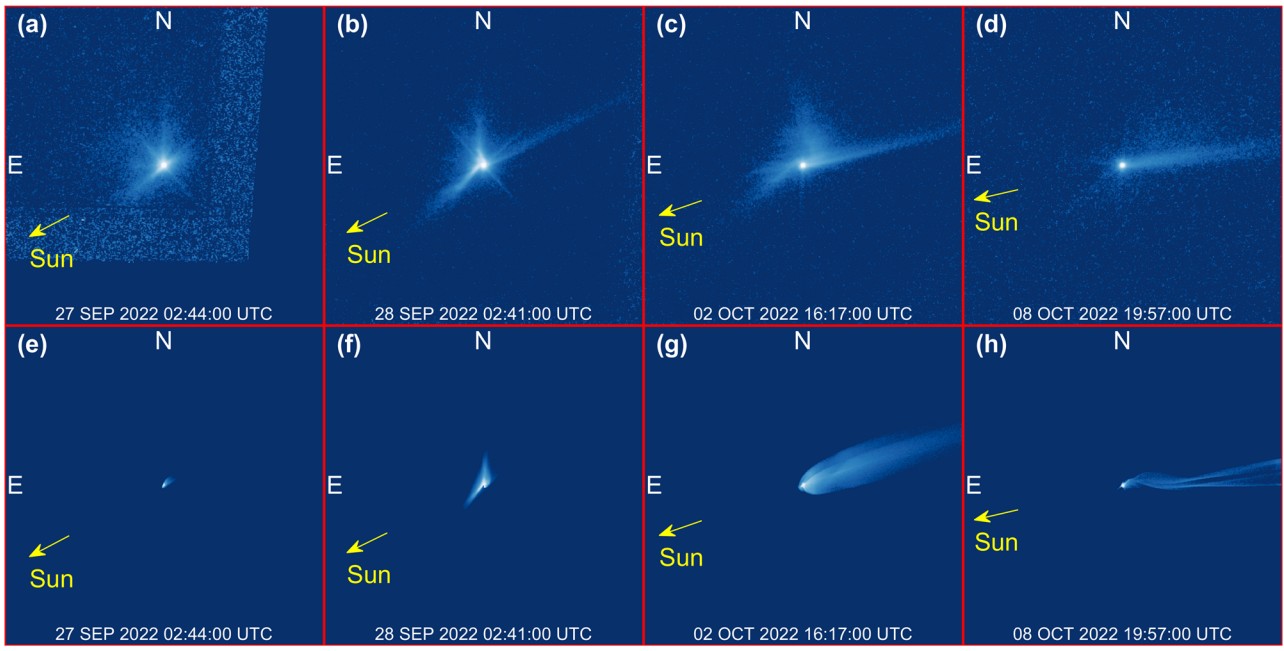

**Fig. 5 | Examples of rejected simulations due to unmatched dynamical conditions.** **e**–**h** Show synthetic images associated to HST reference images in (**a**–**d**) respectively. Each frame is 2160 km wide. With reference to Supplementary Table 1, **e** is associated with simulation set ID 16, **f** with simulation ID 07, **g** with simulation set ID 03, and **h** with simulation set ID 21. The mismatch between synthetic and HST images is used to constrain the properties of the ejecta.

direction. The oscillations help constraining the mass of ejecta, providing a lower-end value, as the curvature is visible to HST only above a critical mass of ~5 × 10⁶ kg. The timing and geometry of the bifurcation depends on the VSD of the ejecta, but no general trends are observed in the simulation set.

### Mass and size-frequency distribution of the ejecta

Admissible ranges of ejecta mass and particle size distribution are assessed by quantifying the error between the photometric flux of HST images and the simulated synthetic ones, for a range of values of masses and dSFD coefficients (see "Methods" Eq. 4). Figure 6 reports the flux error as a function of the mass and dSFD coefficient for two selected sets of simulations, which represent our best-fit for the cone-related and the tail-related features. The colorbar in Fig. 6 is restricted to maximum values reported in each figure. The flux error in the yellow region is equal or larger than these maximum values.

The properties of the cone-related features are well constrained by HST observations as the photometric error has a clear localized set of minima (Fig. 6b). We find they require a mass in the range $1$–$5 \times 10^6$ kg depending on the dSFD coefficient, which ranges between −3 and −2.7. Beyond this range, the photometric error grows rapidly up to several tens of W/(m² μm) (yellow region), and no solutions are found

for mass values above $1.5 \times 10^7$ kg as and dSFD coefficient lower than −3.3, as well as for a mass lower than $10^6$ kg, for which the cone-related features are not visible in the rendered images.

The properties of tail-related features are harder to constrain, as the total mass of ejecta contributing to the tail includes ejecta that are still close to Didymos after 14 days, which are not visible in HST images, and therefore unconstrained by observations. In our simulations, a large fraction of the mass remains in the system, suggesting that more realistic estimates of the dSFD coefficient and the total ejecta mass would require additional information/constraints about inner ejecta. In fact, an increase in the total mass ejecta produces feasible solutions with higher dSFD coefficient, and higher mass orbiting within the inner system. In this context, we can constrain lower bounds, as below an ejecta mass of $10^7$ kg the tail-related features are not visible in the simulated HST images as the photometric error is close to the absolute value of the flux contained in the image. We also identify regions in the mass-dSFD-coefficient plane where the photometric error increases rapidly with ejecta radiometric flux up to several tens of W/(m² μm) (yellow region in Fig. 6a) providing upper bounds to the mass of ejecta. For the best-fit case presented in this work, we selected an dSFD coefficient of −2.4 (ref. 9), which leads to a mass of $2.5 \times 10^7$ kg, including rapid re-accretions (~$8.4 \times 10^6$ kg). The tail profile is also

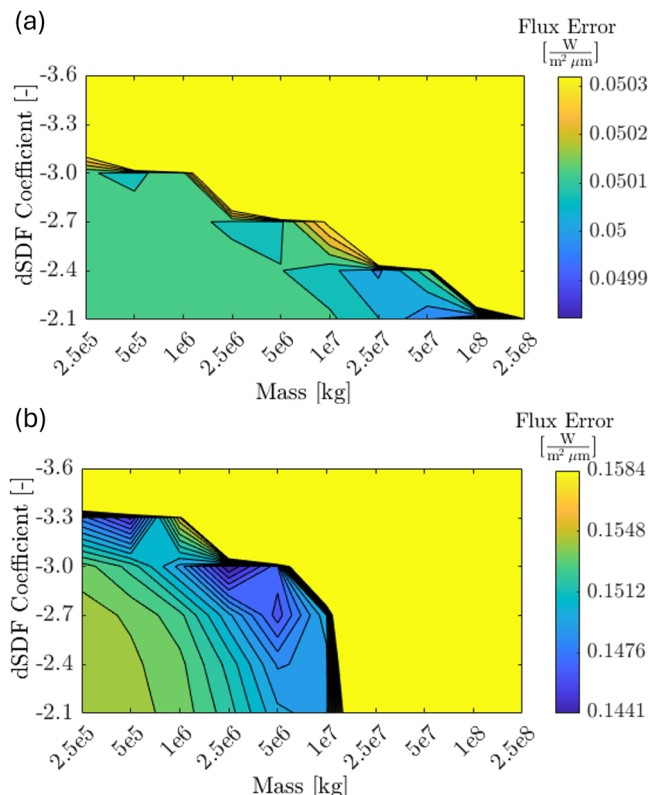

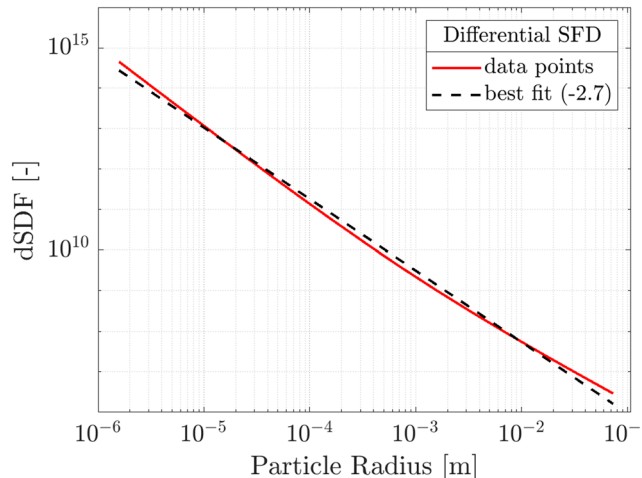

**Fig. 7 | Differential SFD.** Computed combining all ejecta particles involved in cone-related and tail-related features, and its best-fit value −2.7. Data refers to simulation set IDs 19 and 02 (see Supplementary Table 1). Source data are provided as a Source Data file.

**Fig. 6 | Photometric error versus mass and dSFD power-law coefficient.** Simulations forming **a** the tail-related feature and **b** the cone-related features. We identify a mass in the range $1–5 \times 10^7$ kg with dSFD coefficient of −2.4 to produce tail-related features, and in the range $1–5 \times 10^6$ kg with dSFD coefficients between −3 and −2.7 to produce the cone-related features. The colorbar is restricted to maximum values reported in each figure. The error in the yellow region is equal or larger than these maximum values. Figures refer to simulation set IDs **a** 19 and **b** 02 (see Supplementary Table 1). Source data are provided as a Source Data file.

consistent with higher values of mass in case the particle size distribution is flatter as the dSFD coefficient increases.

When considering the overall ejecta population, the overall mass is found to be in the range $1.1–5.5 \times 10^7$ kg, with a best-fit value of $1.9 \times 10^7$ kg. We remark this is a lower-bound, due to the unknown population of larger, unconstrained ejecta particles that remain bounded to the Didymos system. We find this is consistent with studies based on close-proximity LICIACube observations[20,21]. Figure 7 shows the dSFD obtained by combining ejecta particles involved in the cone-related and the tail related features, and its best fit resulting into a value of −2.7.

## Implications

Dynamical mechanisms such as spiral feature, curved ejecta streams, curved tail and tail bifurcation arise naturally from the interaction between the gravitational environment of a binary system and SRP. Although other mechanisms are consistent with the formation of such features, we show that natural binary dynamics is a viable mechanism. This implies that when detected, features might be used to investigate the binary nature of an asteroid, or e.g., to constrain the orbit of a secondary, by measuring the time between oscillations. This applies to both active asteroids and to impact-driven events[36]. Accordingly, discontinuities in the evolution of active objects[43–46] might be diagnostic of dynamical processes undergoing within the asteroid, e.g., breakup, fission, and formation of binaries, without the need for additional ejecta-producing events[33–35]. For impact events, such features might reveal the formation of a secondary out of bounded ejecta, for asteroids that do not previously have a secondary.

For the case of Didymos, the significant quantity of bound ejecta having close encounters and collisions with Dimorphos could change its orbit period slightly, due to a small change of its mass and inertia properties[38,47–49]. This is consistent with the steady change of orbital period observed in the months after the DART impact[5]. The abundance of ejecta reaccreted on the surface of Didymos and Dimorphos will be observed by ESA's Hera spacecraft, which will visit the Didymos system in 2027[50]. This will provide new data to constrain to the upper bound value of the inner ejecta mass, which is the most relevant unconstrained parameter in our analysis.

The amount of ejecta mass created after the impact has implications on the impact process itself, and therefore on the properties of Dimorphos. Our $1.9 \times 10^7$ kg estimate is equivalent to ~0.4% of the mass of Dimorphos, assuming a bulk density of 2400 kg/m³, and can go up to ~1–2% in case the amount of ejecta in the inner system is considerably larger. We find that a large fraction of ejecta mass never escapes Dimorphos and is quickly reaccreted. This is not accounted in our total ejecta mass estimate, and accounts to ~$8.4 \times 10^6$ kg. These numbers are consistent with a very weak, rubble-pile Dimorphos, with very low cohesion between its constituents[4,19].

## Methods
### Methodology overview

A schematic overview of the methodology is reported in Supplementary Fig. 3. More details on data and functional blocks reported in the Figure, as well as the full simulation/processing pipeline are provided in the next paragraphs. Numerical tools used to propagate the dynamics of the ejecta and for the analysis of images are developed internally at Politecnico di Milano.

The dynamics of ejecta particles are simulated under the gravitational influence of Didymos, Dimorphos and the Sun, and the SRPs. More details are found in the Dynamical simulations subsection. Overall, 30 simulation sets were run, each propagating trajectories of 5 million ejecta particles, using a high-fidelity model of their dynamics and ephemerides of the Sun and asteroids[6]. The range of particle radii was selected after some preliminary simulations, to ensure it contains ejecta involved in the formation of the features reported below. This was needed to avoid severe computational burdens, while still considering a representative subset of ejecta that can be compared with observations. Due to the impossibility to simulate all particles involved, we use an image multiplication procedure to compensate for this issue[51]. The multiplicative compensation procedure, as detailed in

its dedicated subsection, weights each simulated ejecta particle as a cluster of particles according to the total mass and the dSFD coefficient. The multiplicative compensation procedure output is a series of image multipliers which modifies the radiative flux of each HST pixel according to the particle size and the ejecta total mass. Indeed, we consider these two parameters as the ejecta characteristics to be determined with a parametric analysis reported hereafter. More details about the multiplicative compensation procedure are reported below.

From these inputs, we performed a preliminary investigation to identify the most suitable set of simulations in terms of ejecta dynamical parameters (i.e., the ejecta velocity-size distribution). This investigation has been performed by rendering images with coarse values of ejecta total mass (i.e., $10^6$ kg and $10^7$ kg) and one dSFD coefficient (i.e., $-2.7$) and by comparing visually the cone-related and tail-related features between real and synthetic images. The synthetic rendered images are generated by emulating the observational geometry of the HST and by assuming the photometrical ejecta characteristics from the literature[51–54]. More details about the rendering procedure are reported below. We performed a preliminary selection of simulations focusing on the timing of features and their appearance by visual inspection. We identified that the tail- and the cone-related features are associated with different velocity ranges. Examples of discarded simulations are reported in Fig. 5. We use the selected dynamical parameters to simulate a dataset of synthetic images by varying over a finer grid of total ejecta mass and dSFD coefficients, generating a large dataset of synthetic images to be compared against HST real images. We perform this comparison by computing the radiometric flux error. The radiometric flux error is defined as the sum of the error between the flux in each pixel of the simulated image and the HST image over all the considered pixels. As we are interested in only some specific region of the HST images, we perform this sum only on some areas defined by a coarse label identifying three main regions in the HST image: the noise mask, the cone-related mask, and the tail-related mask. More details about each mask are provided in the following paragraphs. The cone-related and tail-related masks are used to compute the radiometric flux error for the associated features in a neighbor of the HST feature location. The noise mask is used to estimate the mean level of image noise. This provides a threshold to the radiometric flux, and values below this threshold do not contribute to the photometric error. We perform this step as the rendered images are noiseless and we need to avoid overfitting of the rendered data with HST image noise. This procedure is used to compute numerical values reported Fig. 6, which in turn is used to select the total ejecta mass and dSFD coefficient for both the cone-related feature and the tail-related feature.

Finally, we render a final image merging the cone-related and tail-related velocity distributions considering its respective the total ejecta mass and its respective dSFD coefficient. These rendered images are the ones shown in Figs. 2 and 4. We use these final images to extract the best-fit simulation data reported in Table 1 by using a set of high-fidelity masks. The high-fidelity masks are computed by labeling HST images to identify detailed areas where each feature is located. More details about these masks and the labeling are reported below.

## Label generation

The main goal of this step is to generate two sets of labels: the coarse masks and the Hi-Fi masks. In the text we will refer to image labels or image masks as synonyms. These two image masks are used for different applications. The formers are exploited to compute the radiometric flux error, while the latter are used to compute the ejecta morphological characteristics reported in Table 1.

To gather the image labels, the set of 16 short-exposure images acquired by HST after the impact is used to extract the masks. The features are identified manually on the images using the "Image Labeler" tool by Matlab. Once identified, the annotated labels act as a mask, that can be applied logically to the projected simulation results to quantify physical properties associated with them.

For the coarse labeling step, three main masks are identified in HST images:
1. The noise mask is the image region where no information about the cone-related and tail-related features is present. Therefore, this area is used to gather the mean image error to compute the radiometric flux error
2. The tail-related mask which contains the information linked with the tail
3. The cone-related mask which contains the information linked with the ejecta spiraling behavior

Examples of the coarse labels are shown in Fig. 8 along with the real HST image. Note that the cone-related and the tail-related labels include not only the area where the associated feature are present, but also a region of the image where this feature is not present. This is done to penalize simulations where the ejecta in the simulated images are more spread around with respect to the real HST images. Indeed, when this occurs, the radiometric flux error increases avoiding converging to a wrong solution.

For the high-fidelity labeling step, five main masks are labeled which are associated with five main features:
1. The central blob of pixels which is the brightest area in the image. This image region is associated with the Didymos system
2. The spiral North front which is the portion of the cone-related features ejected from Dimorphos in the direction of Didymos (i.e., the North as projected in the HST image)
3. The spiral South front which is the portion of the cone-related features ejected from Dimorphos in the direction opposite of Didymos (i.e., the South as projected in the HST image)
4. The tail which is composed of both the primary and secondary tail (when present)
5. The background which is defined as the image region complementary to all other labels.

Figure 9 shows examples of the high-fidelity labels along with the prelabeled HST images. These labels are used to postprocess the final rendered image to compute morphological characteristics as the ones reported in Table 1.

## Dynamical simulations

We assume that Didymos and Dimorphos do not feel the gravitational effects of the ejecta, although a significant quantity of bound ejecta could subtly affect the binary orbit period[38], as mentioned above. However, this is not relevant for the dynamical evolution of DART ejecta features. To perform dynamical simulations, we use a quasi-inertial reference frame centered at the binary system barycenter. Axes are inertially fixed, with X and Y lying on the ecliptic plane at epoch J2000, and Z completing the orthogonal triad. The use of prefix quasi highlights how the system should be considered inertial for characteristic times shorter than the Didymos heliocentric revolution period[30]. The dynamics of particles are driven by the central gravity field of Didymos and Dimorphos, the third-body effect of the Sun, and the SRP. Celestial bodies positions are resolved precisely, as provided by high-order integration in NASA's DART mission kernels[6]. In our simulations, particles are assumed to be homogeneous and of spherical shape. Their mutual gravitational interactions are neglected. The dynamical propagator was developed at Politecnico di Milano and is coded in MatLab.

The equations of motion for a point mass (i.e., a particle) in the proximity of Didymos system read[55]

$$\ddot{\mathbf{r}} = -\mu_{D1} \frac{\mathbf{r} - \mathbf{r}_{D1}}{||\mathbf{r} - \mathbf{r}_{D1}||^3} - \mu_{D2} \frac{\mathbf{r} - \mathbf{r}_{D2}}{||\mathbf{r} - \mathbf{r}_{D2}||^3} - \mu_S \left( \frac{\mathbf{r}_S}{||\mathbf{r}_S||^3} + (1 - \beta) \frac{\mathbf{r} - \mathbf{r}_S}{||\mathbf{r} - \mathbf{r}_S||^3} \right)$$

$$(1)$$

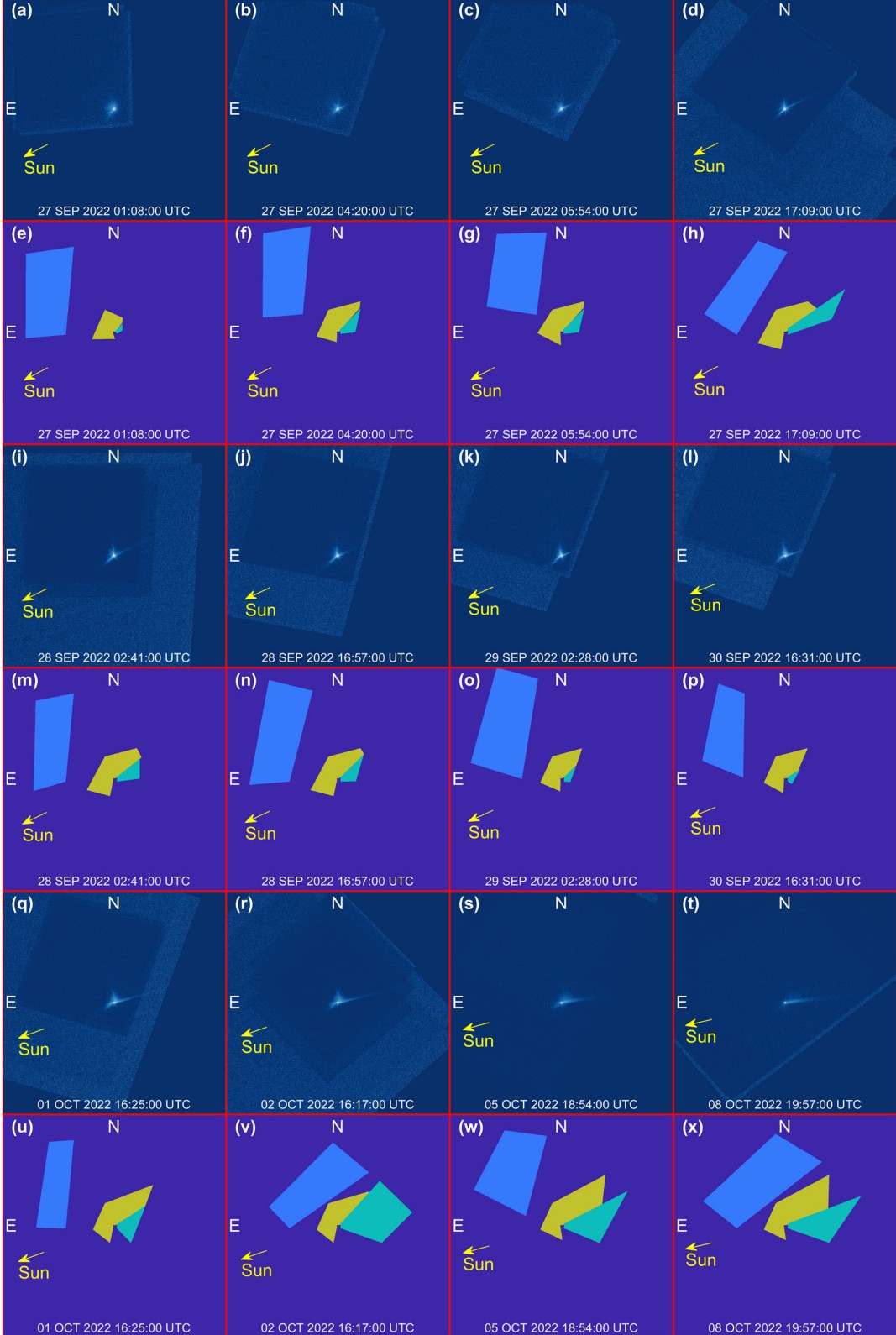

**Fig. 8 | Coarse identification and labeling of features.** **e–h**, **m–p**, **u–x** Report noise masks in light blue, cone-related masks in yellow, and tail-related masks in green. Associated HST images are reported in (**a–d**, **i–l**, **q–t**). Each frame is 8640 km wide.

Where $\mu_{D1} = 3.490 \times 10^{-8}$ km³s⁻², $\mu_{D2} = 3.246 \times 10^{-10}$ km³s⁻², and $\mu_S = 1.327 \times 10^{11}$ km³s⁻² (refs. 5,38) are the gravitational parameters of Didymos, Dimorphos, and the Sun, respectively; $\mathbf{r}$, $\mathbf{r}_{D1}$, $\mathbf{r}_{D2}$, and $\mathbf{r}_S$ are the position vectors of the particle, Didymos, Dimorphos, and the Sun

with respect to the binary system barycenter, respectively. Lastly, the ballistic coefficient $\beta = P_0 D_{Au}^2 C_r^* / (c\mu_S)$ where[56] $P_0 = 1367$ Wm⁻² is the solar flux at 1 AU; $D_{AU} = 1.495 \times 10^8$ km is the Sun–Earth distance; $c = 2.998 \times 10^8$ ms⁻¹ is the speed of light in vacuum;

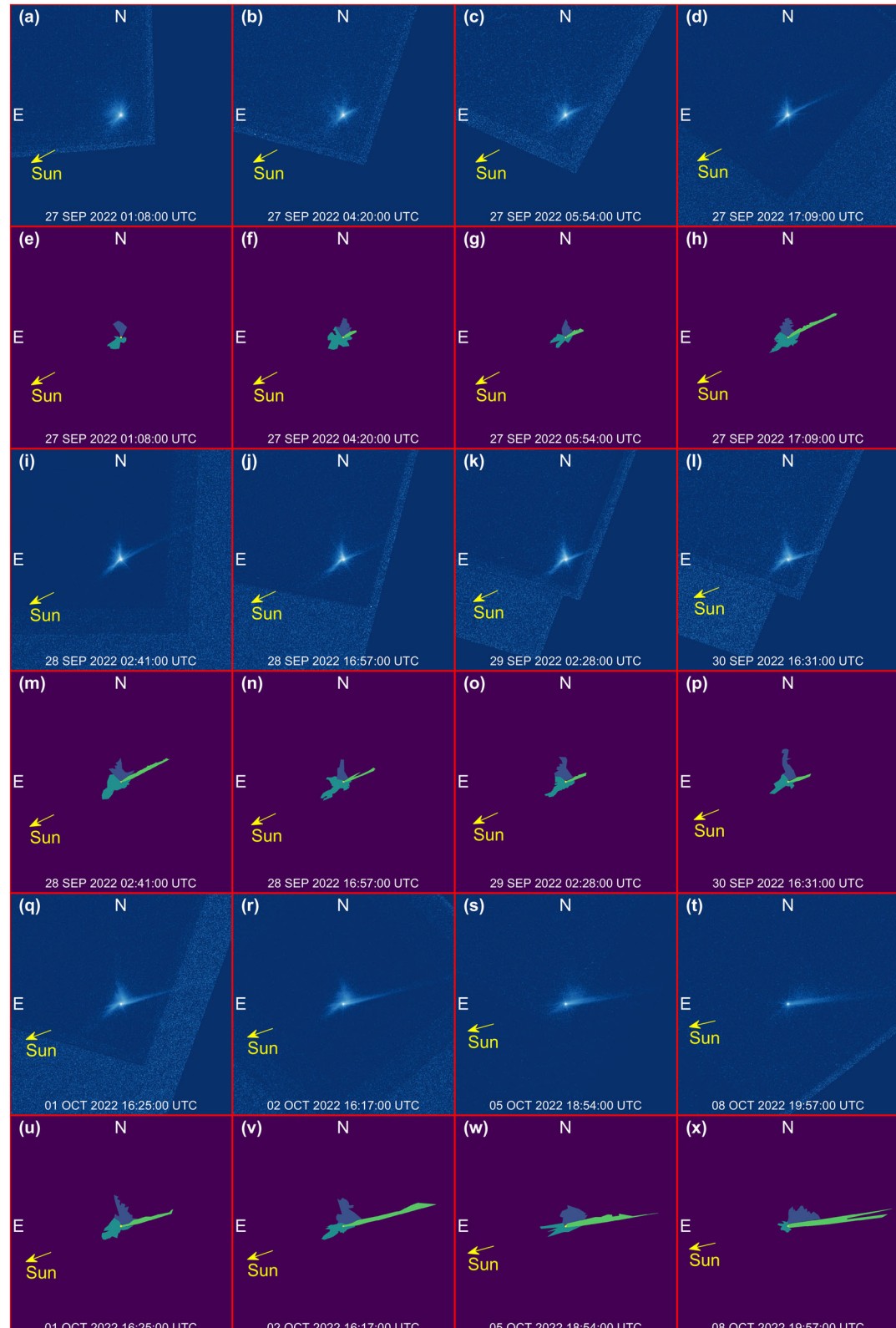

**Fig. 9 | High-fidelity identification and labeling of features. e–h, m–p, u–x** Reports the identification of the main ejecta features: northern (blue) and southern (dark green) arms of the spiral, tail (light green) and Didymos system (yellow). Associated HST images are reported in (**a–d**, **i–l**, **q–t**). Each frame is 4752 km wide.

$C_r^* = C_r A/m = 3C_r/(4R\rho)$ with $C_r$ the reflectivity coefficient, $A$ the equivalent surface area, $m$ the mass, R the radius, and $\rho$ the density of the particle, set to 3.5 g/cm$^3$ to match the grain density of L and LL chondrites[57], which are the best meteoric analogs for Didymos[58,59]. The equations of motion are integrated with a multistep, variable-step,

variable-order, Adams–Bashforth–Moulton, predictor-corrector solver of orders 1st–13th[60]. The dynamics are propagated with relative and absolute tolerances both set to $10^{-9}$.

We propagate several sets of initial conditions. Each set contains $N_p = 10^6$ particles for each of the $N_R = 5$ size distribution ranges

$[R_i, R_{i+1}]$ with $R_i = 10^{-6+i}$ meter for $i = 0, \ldots, 4$. A certain fraction of the particles are assumed to be isotropic ejecta, while the remaining fraction are arranged in a cone. Several isotropic/cone partition levels have been tested. However, isotropic particles were finally removed as their contribution was not in agreement with HST observations.

Each particle initial condition is constructed as follows:

i.  sampling the particle radius $R$ from $\mathscr{U}(R_i, R_{i+1})$, where $\mathscr{U}(a, b)$ is the uniform distribution bounded in $[a, b]$;

ii. assigning the DART spacecraft impact location as the initial position $\mathbf{r}_0$;

iii. computing the magnitude of the initial velocity $v_0 = \|\mathbf{v}_0\|$ at the impact epoch $t_0$ for a particle belonging to:
   a. the isotropic contribution as a random velocity sampled from $\mathscr{U}(v_{esc}, 2v_{esc})$, where $v_{esc}$ is the Didymos escape speed from the impact point;
   b. the cone as the sum of the Didymos escape speed from the impact point $v_{esc}$ and a term dependent on the particle radius $v_{radius}$:

$$v = v_{esc} + \varrho v_{radius} \qquad (2)$$

   where $\varrho$ is a multiplicative coefficient sampled from the uniform distribution $\mathscr{U}(0, 1)$ and $v_{radius}$ is sampled from the velocity-size distribution (VSD) given by:

$$\pi_v(r) = \kappa r^{-\gamma} \qquad (3)$$

   where $\kappa$ and $\gamma$ are the coefficient and exponent, respectively, of velocity-size distribution, and $r$ is the particle radius in meters.

iv. calculating the initial velocity direction $\hat{\mathbf{v}}_0 = \mathbf{v}_0/v_0$ for particles belonging to:
   a. the isotropic contribution as $\hat{\mathbf{v}}_0 = \left[\sqrt{1-z^2}\cos\phi, \sqrt{1-z^2}\sin\phi, z\right]^\top$ with $z$ and $\phi$ sampled from distributions $\mathscr{U}(0, 1)$ and $\mathscr{U}(0, 2\pi)$, respectively, such that it belongs to the hemisphere facing the ejecta release direction and having the pole aligned with the DART spacecraft velocity at the impact epoch $t_0$;
   b. the cone as $\hat{\mathbf{v}}_0 = [\sqrt{1-z^2}\cos\phi, \sqrt{1-z^2}\sin\phi, z]^\top$ with $z$ and $\phi$ sampled from distributions $\mathscr{U}(\cos\theta_1, \cos\theta_2)$ and $\mathscr{U}(0, 2\pi)$, respectively, such that it is bounded between two coaxial cones with half-angles $\theta_1 = 60$ deg and $\theta_2 = 70$ deg, and axis aligned with the DART spacecraft velocity at the impact epoch $t_0$;

v. converting the initial position $\mathbf{r}_0$ and velocity $\mathbf{v}_0$ from the frame centered at Dimorphos to the frame centered at the binary system barycenter.

Supplementary Table 1 summarizes the velocity scaling power law $(\kappa, \gamma)$ pairs selected to sample the random contribution $v_{rand}$ of the particle initial velocity, while a plot of the VSD is reported in Supplementary Fig. 1. The maximum velocity (corresponding to the minimum radius $R = 10^{-6}$ m) of the distribution is reported as well.

**Multiplicative compensation procedure**

As previously mentioned, it is not possible to simulate the number of ejecta particles involved in the simulation, given their large number. We exploited and extended a preexisting methodology developed to simulate a smaller number of particles[51]. This implies the calculation of some multiplicative compensation factors that are used to weight each simulated ejecta particle as a pack of particles according to the total mass and the dSFD coefficient. The main steps of the used methodology are expounded hereunder.

We define $\pi_N(r)$ as the ejecta dSFD between a minimum radius $r_m$ and a maximum radius $r_M$ and normalized in such a way that its integral

is equal to 1. Its expression is:

$$\pi_N(r) = \frac{(1-a)}{r_M^{(1-a)} - r_m^{(1-a)}} r^{-a} \qquad (4)$$

where $r$ is the particle radius and $a$ is the dSFD coefficient. Note that the SFD is the integral of the dSFD.

Let $M$ be the ejecta total mass and $N$ the total number of particles. The total mass can be computed as:

$$
\begin{aligned}
M &= \rho N \int_{r_m}^{r_M} V(r)\pi_N(r)\mathrm{d}r = \rho N \int_{r_m}^{r_M} \frac{4}{3}\pi r^3 \pi_N(r)\mathrm{d}r \\
&= \rho N \frac{4}{3}\pi \frac{(4-a)}{(1-a)} \frac{r_M^{(1-a)} - r_m^{(1-a)}}{r_M^{(4-a)} - r_m^{(4-a)}}
\end{aligned} \qquad (5)
$$

where $V(r)$ is the particle volume as a function of $r$ and $\rho$ is the particle density assumed constant. By assuming the total ejecta mass and the dSFD coefficient, it is possible to derive the total number of particles between $r_m$ and $r_M$:

$$N = \frac{M}{\rho}\frac{3}{4\pi}\frac{(1-a)}{(4-a)}\frac{r_M^{(4-a)} - r_m^{(4-a)}}{r_M^{(1-a)} - r_m^{(1-a)}} \qquad (6)$$

Note that the SFD coefficient $a\prime$ is related with the dSFD coefficient $a$ by $a' = a - 1$. At this stage it is necessary to compute the multiplicative compensation factors which depends on the particle radius. Indeed, particles must be weighted according to the selected dSFD as a single particle represent a pack of particle which are distributed according to $\pi_N(r)$. To ease the calculations, we divide the interval $[r_m, r_M]$ in subintervals. Let $r_{l_i}$ and $r_{u_i}$ be the lower and upper bound of the $i$th bin. The fraction of particles within the $i$th bin is labeled $n_i$ and it is computed as:

$$n_i = \int_{r_{l_i}}^{r_{u_i}} \pi_N(r)\mathrm{d}r = \frac{r_{u_i}^{(1-a)} - r_{l_i}^{(1-a)}}{r_M^{(1-a)} - r_m^{(1-a)}} \qquad (7)$$

Let $N_{\mathrm{sim}_i}$ be the number of effective particles simulated within the $i$th bin. It is straightforward to compute the multiplicative compensation factor $k_i$ each ejecta particle has so to match the desired fraction of particle in that bin with a given total ejecta mass and dSFD coefficient:

$$k_i = \frac{N n_i}{N_{\mathrm{sim}_i}} \qquad (8)$$

Note that this multiplicative compensation factor is constant within the same bin, implying that particles with different radii are weighted in the same manner. Moreover, we used the mean particle radius within the considered bin to render ejecta particles avoid small faint particle which are weighted as larger bright particles. To average the behavior of the particle within the same bin, we decided to add a second computational step to infer the multiplicative compensation factor $k_i$. We use the fraction of particles within the $i$th bin $n_i$ to define a piecewise constant function $\bar{\pi}_N(r)$ representing the dSFD defined between $r_m$ and $r_M$ and normalized in such a way that its integral its equal to 1. This function is constant within the bin defined by $r_{l_i}$ and $r_{u_i}$ and it contains $n_i$ percentage of the total particle. It can be expressed as follows:

$$\bar{\pi}_N(r) = \sum_{i=1}^{n_{\mathrm{bins}}} \frac{n_i}{r_{u_i} - r_{l_i}} \mathbb{I}(r_{l_i} \le r \le r_{u_i}) \qquad (9)$$

where $\mathbb{I}(\bullet)$ is the indicator function, defined as 1 in the input interval; 0 otherwise.

The total ejecta mass is:

$$M = \rho \bar{N} \frac{\pi}{3} \sum_{i=1}^{n_{\text{bins}}} n_i \left( r_{u_i}^2 + r_{l_i}^2 \right) \left( r_{u_i} + r_{l_i} \right) \quad (10)$$

where $\bar{N}$ is the total number of particles associated with the distribution $\bar{\pi}_N(r)$. Therefore, $\bar{N}$ is computed as:

$$\bar{N} = \frac{M}{\rho} \frac{3}{\pi} \frac{1}{\sum_{i=1}^{n_{\text{bins}}} n_i \left( r_{u_i}^2 + r_{l_i}^2 \right) \left( r_{u_i} + r_{l_i} \right)} \quad (11)$$

The multiplicative compensation factor $\bar{k}_i$ associated with the distribution $\bar{\pi}_N(r)$ are computed as:

$$\bar{k}_i = \frac{\bar{N} n_i}{N_{\text{sim}_i}} \quad (12)$$

These are the multiplicative compensation factors we use to weight each particle in the simulations according to the bin where it belongs to.

### Synthetic image generation

To perform synthetic image generation, we simulate the HST with a pinhole camera model[61] with focal length of 57.6 m, an iFoV (instantaneous Field of View) of 0.194 μrad in both directions and resolution of 2001 pixels in both directions[51]. The first step we apply is to compute which particles fall within the HST field of view to reduce the calculation in projecting them in the image plane. Then we project all the filtered particle on the image plane to identify which pixel is associated with the considered particle. Per each particle we compute the ejecta particle visual magnitude at HST observational geometry $m_{\text{particle}}$:

$$m_{\text{particle}} = m_{1,0} + 5 \log_{10} \left( d_{P,C} \, d_{P,S} \right) + G(\alpha) \quad (13)$$

where $m_{1,0}$ is the particle magnitude at 1 AU from Sun and observer with zero phase angle, $d_{P,C}$ is the distance between the particle and the camera in AU, $d_{P,S}$ is the distance between the particle and the Sun in AU, and $G(\alpha)$ is a phase function correcting for phase angle $\alpha$. We use the following phase function[52]:

$$G(\alpha) = 0.013\alpha \quad (14)$$

where $\alpha$ is expressed in degree. Moreover, we compute the magnitude of the particle at 1 AU with zero phase angle as[51,53]:

$$m_{1,0} = 5 \log_{10} \left( \frac{1329}{2 \, 10^{-6} \, r_{\text{particle}} \sqrt{a}} \right) \quad (15)$$

where $r_{\text{particle}}$ is the particle radius expressed in mm and $a$ is the particle albedo assumed 0.15. Note that we divide the considered range of particle radius in several bins to ease the computation of the multiplicative compensation factor $\bar{k}_i$. As all particles are weighted equally in the same bin, we use the mean particle radius of the bin to avoid giving more importance to larger particle in the same bin.

We compute the radiometric flux $E$ by using Vega reference parameters:

$$E_{\text{particle}} = E_{\text{Vega}} 10^{\left( \frac{m_{\text{Vega}} - m_{\text{particle}}}{2.5} \right)} \quad (16)$$

where $m_{\text{Vega}} = 0.03$ is the Vega magnitude, $E_{\text{Vega}} = 3.631 \times 10^{-8}$ W/(m$^2$ nm) is Vega reference irradiance in the visible. At this stage we sum all radiometric flux contributions in the same pixel coming from the particles in the same bin and we multiply these values for the multiplicative compensation factor $\bar{k}_i$. Moreover, we sum up all the contributions coming from different bins to obtain the image for all the simulated ejecta. Finally, to consider the HST WFC3 F350LP filter and obtain images in the same units of HST data, we perform the following calculations:

$$I_{\text{final}} = \frac{I}{\text{iFoV}^2} \frac{E_{\text{Filter}}}{E_{\text{Vega}}} \quad (17)$$

where $I$ is the rendered image, $I_{\text{final}}$ is the final image coherent with HST data, and $E_{\text{Filter}} = 2.7554 \times 10^{-8}$ W/(m$^2$ nm) is the F350LP filter zero point.

## Data availability

Source data are provided with this paper. All raw HST data associated with this Article are archived and are publicly available at the Mikulski Archive for Space Telescopes (https://mast.stsci.edu/search/ui/#/hst/results?proposal_id=16674) hosted by the Space Telescope Science Institute. The numerical simulation data, synthetic images, masks, and labels generated in this study have been deposited in the Zenodo database and are publicly available at https://doi.org/10.5281/zenodo.14630436. Source data are provided with this paper.

## Code availability

The code used to perform the analysis reported in this work is available from the corresponding author upon request.

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

## Acknowledgements

This work was supported by the DART mission, NASA Contract 80MSFC20D0004. F.F. acknowledges support from the European Union ERC Grant agreement No. 101077758 (TRACES). Views and opinions expressed are however those of the authors only and do not necessarily reflect those of the European Union or the European Research Council. Neither the European Union nor the granting authority can be held responsible for them. F.F. acknowledges support from the Swiss National Science Foundation (SNSF) Ambizione grant No. 193346. J.-Y.L. acknowledges the support provided by NASA through grant HST-GO-16674 from the Space Telescope Science Institute, which is operated by the Association of Universities for Research in Astronomy, under NASA contract NAS 5-26555. The work of E.G.F. was carried out at the Jet Propulsion Laboratory, California Institute of Technology, under a contract with the National Aeronautics and Space Administration (#80NM0018D0004). S.D.R. and M.J. acknowledge support by the Swiss National Science Foundation (project number 200021 207359). S.S. acknowledges financial support from grant ref MR/W009498/1 of the UK Research and Innovation. P.M. acknowledges the French space agency CNES, ESA and The University of Tokyo. F.M. acknowledges financial support from grants PID2021-123370OB-I00 and CEX2021-

001131-S funded by MCIN/AEI/10.13039/501100011033. G.T. acknowledges financial support from project FCE-1-2019-1-156451 of the Agencia Nacional de Investigacíon e Innovacíon ANII and Grupos I + D 2022 CSIC-Udelar (Uruguay). J.O. acknowledges support by grant PID2021-125883NB-C22 by the Spanish Ministry of Science and Innovation/State Agency of Research MCIN/AEI/ 10.13039/501100011033 and by "ERDF A way of making Europe". J.O. and I.H. acknowledge support by the Spanish Research Council (CSIC) support for international cooperation: I-LINK project ILINK22061. O.K. acknowledges funding support from the PRODEX program managed by the European Space Agency (ESA) with help of the Belgian Science Policy Office (BELSPO). A.C.B. acknowledges funding by the NEO-MAPP project 717 GA 870377, EC H2020-SPACE-718 2018-2020/H2020-SPACE-2019, and by MICINN (Spain) PGC2021, PID2021-125883NB-C21. S.I., A.L., M.P., A.R. and F.T. acknowledge support by the Italian Space Agency (ASI) within the LICIACube project (ASI-INAF agreement n. 2019-31-HH.0) and HERA project (ASI-INAF agreement n. 2022–8-HH.0). The simulation campaign was performed using the Euler supercomputer at DAER/PoliMi, the authors thank Francesco Topputo for the support and discussions.

## Author contributions

F.F. conceived the work and led the study, the interpretation of results, and the writing of the manuscript. P.P., G.M., C.G., and M. Pugliatti contributed to the interpretation and analysis of results and writing of the manuscript. P.P. and M. Pugliatti developed techniques for photometric analysis, feature identification, and synthetic image generation. G.M., C.G., and F.F. developed the dynamical propagator and ran the simulation campaign. J.-Y.L. contributed to obtaining and analyzing HST data. J.-Y.L., E.G.F., S.S., M.H., F.M., G.T., H.A., Y.Z., A.R., and D.J.S contributed to the discussion and interpretation of the dynamics of ejecta features. J.M.S., C.C.M., P.M., and O.K. contributed to the discussion about implications for evolutionary processes in binary and active asteroids. S.D.R. and M.J. contributed to the discussion about implications for Dimorphos. J.O. and I.H. contributed with the discussion about inhomogeneous structures in the ejecta cone (Supplementary Materials). N.L.C., A.F.C., D.C.R., A.S.R., A.C.B., T.L.F., S.I., A.L., M. Pajola, and F.T. contributed substantively to revise the manuscript.

## Competing interests

The authors declare no competing interests.

## Additional information

¹Department of Aerospace Science and Technology, Politecnico di Milano, Milan, Italy. ²Planetary Environmental and Astrobiological Research Laboratory (PEARL), School of Atmospheric Sciences, Sun Yat-sen University, Zhuhai, Guangdong, China. ³Jet Propulsion Laboratory, California Institute of Technology, La Cañada Flintridge, CA, USA. ⁴Space Research and Planetary Sciences, Physikalisches Institut, University of Bern, Bern, Switzerland. ⁵Department of Mechanical and Aerospace Engineering, University of Liverpool, Brownlow Hill, UK. ⁶Georgia Institute of Technology, Atlanta, GA, USA. ⁷Auburn University, Auburn, AL, USA. ⁸Cornell University, Ithaca, NY, USA. ⁹Université Côte d'Azur, Observatoire de la Côte d'Azur, CNRS, Laboratoire Lagrange, Nice, France. ¹⁰Department of Systems Innovation, School of Engineering, The University of Tokyo, Tokyo, Japan. ¹¹Instituto de Astrofísica de Andalucía, CSIC, Granada, Spain. ¹²Departamento de Astronomía, Facultad de Ciencias, Udelar, Uruguay. ¹³University of Maryland, College Park, MD, USA. ¹⁴Department of Astronomy and Department of Geology, University of Maryland, College Park, MD, USA. ¹⁵Centro de Astrobiologia (CAB), CSIC-INTA, Torrejon de Ardoz, Spain. ¹⁶Royal Observatory of Belgium, Brussels, Belgium. ¹⁷Department of Climate and Space Sciences and Engineering, University of Michigan, Ann Arbor, MI, USA. ¹⁸Johns Hopkins University Applied Physics Laboratory, Laurel, MD, USA. ¹⁹IUFACyT—DFISTS. Universidad de Alicante, Alicante, Spain. ²⁰INAF—Osservatorio Astronomico di Trieste, Trieste, Italy. ²¹INAF-OAPD, Astronomical Observatory of Padova, Padova, Italy. ²²IFAC-CNR, Sesto Fiorentino, Italy. ²³University of Colorado Boulder, Boulder, CO, USA. ✉e-mail: fabio1.ferrari@polimi.it

