## [Transparent Peer Review file · Nature Communications]

Morphology of ejecta features from the impact on asteroid Dimorphos

Corresponding Author: Professor Fabio Ferrari

Version 0:

Reviewer comments:

Reviewer #1

(Remarks to the Author)

This study provides an insightful analysis of the dynamics and evolution of ejected material within a binary asteroid, resulting from NASA's DART impact on the asteroid Dimorphos. The authors successfully interpret the unique features observed, such as curved ejecta streams and a tail bifurcation, as a natural outcome of the dynamical interaction of the ejecta with the binary system and solar radiation pressure.

This paper presents the most successful model of DART ejecta to date among several models that have been conducted. The model provides a successful interpretation of the dynamical evolution of ejecta gravitational interacting with Didymos and Dimorphos, offering important implications for future research on active asteroids and thus making a significant contribution to the field.

However, the claim that features such as spirals, curved ejecta streams, curved tail and tail bifurcation can be used to prove that an object is a binary system when detected in other objects requires additional evidence. The statement that the orbit of a secondary can be constrained by measuring the curvature in the tail, which quantifies oscillatory motion, also requires supplementation.

There could be several counterexamples. For instance, a double tail might be due to a secondary event. Curved features could be a result of an asymmetry in the ejection pattern, without having to be a binary system. Also, the binary main-belt comet 288P has not shown any of the binary-related features mentioned in this study.

Once these claims are supplemented and the other comments below are addressed, I would recommend the manuscript for publication in Nature Communications. I also suggest improving the readability of this manuscript.

1. The text mentions that the last panel of Methods Fig. 8 shows the effect of curvature producing an oscillatory motion that propagates in the anti-solar direction, but the figure seems to be missing.
2. In previous models from different studies, a broken power-law size distribution was considered consistent with the observations. However, in this study, a single power law (-2.7) seems to function well. How is the application of a single power law successful within this model? Is this also connected to the binary system's gravitational interaction?
3. The model interprets the secondary tail as tail bifurcation. Previous research supporting a secondary event noted that there was evidence of slight brightening in photometry near the assumed secondary event. A remark on the potential reasons for this brightening could be added.

Reviewer #2

(Remarks to the Author)

"Morphology of ejecta features from the DART impact on Dimorphos and their implications" by Ferrari et al. is an interesting paper describing a detailed analysis of the ejecta outcomes produced by DART

impact experiment on asteroid Dimorphos.

It is certainly worthy of publication but, before that, it requires the inclusion of some clarifications regarding the procedure used for the analysis of HST images.

The method consists in the comparison between HST images and simulated images produced using a multiparameter model of ejecta production and evolution. The comparison is done in terms of the sum of the differences between observed and simulated fluxes in each image pixels, what Authors name "error". The error for different values of mass and SFD coefficient is plotted in the key figure 5.

What is missing is a discussion of the statistical significance of differences of error values related to the random noise in observed image and random fluctuation in the simulation of realistic populations of ejecta, in order to define in a quantitative way the boundaries of the confidence regions in the space of the model parameters.

Something that makes me curious is the large yellow regions in Fig. 5, where value of the error appear constant. Authors mention that in those regions error values are large. I wonder what values the error reaches at the edges of the figure, with respect to the values of the best solutions.

Regarding again fig. 5, Authors say (p. 10): "Figure 5 reports the flux error as a function of the mass and dSFD coefficient for two selected simulations". This sentence is not clear to me: two simulations in what sense? Each point of the plane mass-SFD coefficient represents a simulation.

Concerning the physical model of ejecta initial velocity field, two points should be clarified.

The first one concerns the geometrical assumption. Velocity field is the superposition between a isotropic and a cone component. The isotropic component account for the 10% of particles, while the remaining belong to the cone structure. Where does this partition (10%/90%) come from? Why not 50%/50%, for instance? How do the simulations change by changing these values? The cone component is formed by two coaxial cones with half-angles 60 deg and 70 deg. Also this choice appears poorly argued/justified. What happens with different angles?

The second point concerns the velocity-size distribution (VSD) (eq. 3 at p. 22). It is not completely clear if the values of the parameters k and γ has to be obtained fitting the HST images or, on the contrary, they are assumed a priori. In the first case, which values of k/γ corresponds to the fit best solution? In the second one, how were the values chosen?

Minor remarks:

p. 14 "Overall, 30 simulation sets were run, each propagating trajectories of 5 million ejecta particles in the, using a high-fidelity model ...":
something is missing in the sentence.

p. 26 "indicator function":
maybe for non expert readers it is better to say explicitly what indicator function exactly is (1 inside interval of interest, 0 elsewhere...)

p. 27-28 "We use the following phase function: $G(\alpha) = 0.013 \alpha^{(14)}$ "
Authors quote papers 51, 53 and 54 but it seems to me that the only relevant reference is 52 (Hergenrother et al., 2020). While ref. (53) doesn't seem very related to the choice of the phase function, the other papers (51) and (54) simply refer to the work by Hergenrother et al. (2020).

p. 27 "is the particle magnitude at 1 AU", please add "from Sun and observer" (or something like that).

Reviewer #3

(Remarks to the Author)

In the submitted paper, the authors examine the evolution of impact ejecta created by the DART spacecraft, using numerical simulations of ejected particles, generation of synthetic images and comparison of the results with the images captured by HST.

The presented results are consistent with other works focusing on the DART impact. The process of formation of the ejecta features (an impact cone and a tail created by the SRP) has been described by Li et al. 2023, but to my knowledge it hasn't been studied extensively using numerical simulations. The authors further explain that the tail bifurcation observed in the HST images is a natural consequence of the system's dynamics and does not require secondary impacts, fragment disintegration or any other assumption. This work thus constitutes a valuable contribution and I recommend the paper for publication with the following remarks.

I have two questions for the authors which could be briefly addressed in the paper:

1. Using the synthetic images, would it be possible to further generate a synthetic light curve and compare it with the measured light curve of the Didymos system during the impact? As the measured light curve showed a temporary brightening by a factor 8.3 (Dotto et al. 2024), I'd be curious if a simulation that minimizes the photometric error also showed brightening by a similar factor.

2. The bifurcated tail naturally appears in the synthetic images, but the opening angle is notably larger than in the HST images - while the observed opening angle is around 4° , the synthetic images show an angle closer to 10° . What are some of the variables that can affect this? Could the opening angle be related to some of the simulation parameters? It would be helpful if the measured angle could further constrain some unknown parameters, such as the ejecta SFD, or even the mass of Dimorphos.

Minor issues:

Figure 3: The field of view indicated in the caption is not consistent with the length scale shown in the image. The length scale suggests the field of view is 20km.

Figure 5: The y-axis label should be "dSFD" to avoid ambiguity, especially since the term "SFD" is used in the Methods section for the integral of dSFD. There is further a typo in the b) plot ("SDF").

There are a few typos in the Methods section:

- Typo in Fig. 7 (Ejecta Dynamical Paramteres). The label of the Figure also says "77".
- "... of 5 million ejecta particles in the, using ..."
- "... the cone-related and tail-relate features between ..."
- „Let $\pi N(r)$ the ejecta dSFD defined“
- „its integral its equal to 1.“

Figure 8 uses the term "wing" in the legend, which is not used elsewhere in the paper. It would be better to unify the terminology.

Version 1:

Reviewer comments:

Reviewer #1

(Remarks to the Author)

The authors thoughtfully addressed my comments, and the manuscript is now enhanced and comprehensive. I recommend this manuscript for publication.

Reviewer #2

(Remarks to the Author)

The revised version of the manuscript fully address all my remarks raised about the first version.

I can say, as far as I'm concerned, the paper is ready to be accepted for publication.

Reviewer #3

(Remarks to the Author)

Authors addressed all raised concerns and answered my questions. I'm satisfied with the current state of the manuscript and recommend it for publication.

All the comments pointed out by the Reviewers are addressed on a point-by-point basis, and the changes carried out in the manuscript are clearly indicated. To ease the assessment, the received comments are reported in black, our responses are given in **blue**, the corresponding actions/changes implemented in the manuscript are written in **red**.

We have also updated the information of reference items in the bibliography (submitted, in press, published).

We are grateful to the Editor and Reviewers whose suggestions contributed to improve the overall quality of the paper.

REVIEWER 1

This study provides an insightful analysis of the dynamics and evolution of ejected material within a binary asteroid, resulting from NASA's DART impact on the asteroid Dimorphos. The authors successfully interpret the unique features observed, such as curved ejecta streams and a tail bifurcation, as a natural outcome of the dynamical interaction of the ejecta with the binary system and solar radiation pressure.

This paper presents the most successful model of DART ejecta to date among several models that have been conducted. The model provides a successful interpretation of the dynamical evolution of ejecta gravitational interacting with Didymos and Dimorphos, offering important implications for future research on active asteroids and thus making a significant contribution to the field.

COMMENTS

1. However, the claim that features such as spirals, curved ejecta streams, curved tail and tail bifurcation can be used to prove that an object is a binary system when detected in other objects requires additional evidence. The statement that the orbit of a secondary can be constrained by measuring the curvature in the tail, which quantifies oscillatory motion, also requires supplementation. There could be several counterexamples. For instance, a double tail might be due to a secondary event. Curved features could be a result of an asymmetry in the ejection pattern, without having to be a binary system. Also, the binary main-belt comet 288P has not shown any of the binary-related features mentioned in this study. Once these claims are supplemented and the other comments below are addressed, I would recommend the manuscript for publication in Nature Communications.

We agree that such features are not unequivocally linked to the presence of a binary system. As mentioned by the reviewer, other mechanisms or events may lead to similar morphological structures. Our results demonstrate that ejecta features such as spirals, curved ejecta streams, bifurcations in the tail, may originate from the natural evolution of binary system dynamics alone, without the need for secondary events. However, we do not intend to claim that such natural binary dynamics are the only possible mechanism for originating such features. We clarified this in the text.

We also agree with the fact that relating such features (when detected in other objects) to the binary nature of the system would require additional evidence. Here we only intend to suggest this can be a possibility, providing evidence related to the special case of Didymos. In this case we clearly observe that the oscillatory motion in the tail has the same frequency of the orbit of the secondary. We also remark this is due to particles being ejected from the secondary, i.e., carrying a non-zero orbital velocity with respect to the barycentre of the system. This might not be applicable to e.g., other binary systems where ejection occurs

from the primary. Also, this appears to be related to oscillatory motion, rather than a simple curvature alone. We clarified this in the text (Abstract, Implications).

2. I also suggest improving the readability of this manuscript.

The text was revised carefully by several authors (including English native speakers) to improve its readability.

3. The text mentions that the last panel of Methods Fig. 8 shows the effect of curvature producing an oscillatory motion that propagates in the anti-solar direction, but the figure seems to be missing.

We refer here to panel (h) of Methods Figure 8. Methods Figure 11 was erroneously labelled with number 8. This is now fixed in the text and Figure caption.

4. In previous models from different studies, a broken power-law size distribution was considered consistent with the observations. However, in this study, a single power law (-2.7) seems to function well. How is the application of a single power law successful within this model? Is this also connected to the binary system's gravitational interaction?

We think this is only a matter of a different approach. In our work, we split the contributions of the tail and the cone, showing that they are produced as a result of different dynamical regimes, involving different particle populations (size and velocity). We identify a mass in the range $1-5 \times 10^7$ kg with dSFD coefficient of -2.4 to produce tail-related features, and in the range $1-5 \times 10^6$ kg with dSFD coefficients between -3 and -2.7 to produce the cone-related features. This means we are actually combining together two power laws, with two distinct indices. The single power law arises when considering the overall ejecta population, combining ejecta particles involved in the cone-related and the tail related features: in this case the best fit results in a value of -2.7. Alternatively, one could also try to fit the data using broken laws or other distributions.

5. The model interprets the secondary tail as tail bifurcation. Previous research supporting a secondary event noted that there was evidence of slight brightening in photometry near the assumed secondary event. A remark on the potential reasons for this brightening could be added.

The brightening and the secondary tail might be certainly produced by a secondary event, but this is not the only possible explanation. E.g., Kim & Jewitt (2023) [ref. 42] show that these can be explained by the viewing geometry as well.

From a dynamic perspective, in our work we observe that cm-sized ejecta particles ejected in the Solar direction reverse their velocity due to interaction with SRP and return in the field of view between 8-12 days after the impact, near the observed brightening (Figure 3). We highlighted this in the manuscript. This could be an alternative mechanism to support the observed brightening. However, this is not claimed in the manuscript as it would require a detailed study of the photometric properties of such an event, which is out of the scope of this work. As a matter of fact, the discussion about the brightening and its origin is an important matter, still debated in the scientific community, and therefore should be analysed carefully, possibly in a dedicated work.

REVIEWER 2

The method consists in the comparison between HST images and simulated images produced using a multiparameter model of ejecta production and evolution. The comparison is done in terms of the sum of the differences between observed and simulated fluxes in each image pixels, what Authors name "error". The error for different values of mass and SFD coefficient is plotted in the key figure 5. What is missing is a discussion of the statistical significance of differences of error values related to the random noise in observed image and random fluctuation in the simulation of realistic populations of ejecta, in order to define in a quantitative way the boundaries of the confidence regions in the space of the model parameters.

The comparison between observed and synthetic images is performed only after removing the background random noise. No such noise is present in the simulated images, which contains only particles ejected from the system. Instead of modelling the random noise in HST images and add them to simulated ones, we remove the random noise systematically from each HST image to enable a fair comparison with the simulated images. The noise level/threshold is computed using noise masks, identified systematically in each image (see e.g., Methods Figure 9). This threshold provides the quantitative boundary below which the random errors/fluctuations are not considered. If the overall flux contribution in a pixel results lower then the noise threshold, then this is not considered (the contribution is removed), as the content would not be statistically reliable in that pixel. More details about this procedure are reported in the Methods section (last few paragraphs of "Methodology Overview" subsection, from "We use the selected dynamical parameters..." up to the end of the subsection. This part has been also revised in the manuscript for clarity).

MAJOR COMMENT

1. Something that makes me curious is the large yellow regions in Fig. 5, where value of the error appear constant. Authors mention that in those regions error values are large. I wonder what values the error reaches at the edges of the figure, with respect to the values of the best solutions.

The value of the error in the yellow region is not constant in Figure 5. To enhance the variations in the area where a minimum of the error is present, the colorbar is limited to maximum values reported in each figure. The error in the yellow region is equal or larger than these maximum values. The highest error in the figure is about $85 \text{ W}/(\text{m}^2 \mu\text{m})$ at the figure upper right corner. This has been clarified in the text and in the caption of Figure 5.

2. Regarding again fig. 5, Authors say (p. 10): "Figure 5 reports the flux error as a function of the mass and dSFD coefficient for two selected simulations". This sentence is not clear to me: two simulations in what sense? Each point of the plane mass-SFD coefficient represents a simulation.

That is right, the sentence was unclear. We meant two different sets of simulations. We refer to the tail-related and the cone-related simulation sets, which have been run independently. Each set is a full Monte Carlo set, including several simulations. This has been clarified in the text.

3. Concerning the physical model of ejecta initial velocity field, two points should be clarified. The first one concerns the geometrical assumption. Velocity field is the superposition between a isotropic and a cone component. The isotropic component account for the 10% of particles, while the remaining belong to the cone structure. Where does this partition (10%/90%) come from? Why not 50%/50%, for instance? How do the simulations change by changing these values? The cone component is formed by two coaxial cones with half-angles 60 deg and 70 deg. Also this choice appears poorly argued/justified. What happens with different angles?

The isotropic component is set to zero in the final set of simulations. The 10% reported in the text was misleading: actually, several values have been tested to setup the partition between isotropic and cone distribution. After these tests, we observed that the isotropic distribution would create structures that are never observed in the actual HST images. So we concluded that all ejecta are arranged in a cone structure. We clarified this in the text and removed the 10% to avoid confusion.

The geometry of the cone is set to match the best of current knowledge, using data estimated by LICIACube observations (Dotto et al., 2023) [ref. 21].

4. The second point concerns the velocity-size distribution (VSD) (eq. 3 at p. 22). It is not completely clear if the values of the parameters k and γ has to be obtained fitting the HST images or, on the contrary, they are assumed a priori. In the first case, which values of k/γ corresponds to the fit best solution? In the second one, how were the values chosen?

The properties of the VSD are free parameters for our investigation. We tested several different VSD: their coefficients are chosen a priori and then we select the best fit when comparing to HST images. We chose these coefficients in a range of values to be consistent with known estimates and observations, and are in line with values reported in similar works [e.g., ref. 9, 10, 21, 25, 30]. See Methods Table 2 for an overview of the investigated parameter spaces.

MINOR REMARKS

1. p. 14 "Overall, 30 simulation sets were run, each propagating trajectories of 5 million ejecta particles in the, using a high-fidelity model ...": something is missing in the sentence.

"in the" was a typo, we removed it.

2. p. 26 "indicator function": maybe for non expert readers it is better to say explicitly what indicator function exactly is (1 inside interval of interest, 0 elsewhere...)

We clarified this in the text.

3. p. 27-28 "We use the following phase function: $G(\alpha) = 0.013 \alpha^{14}$ " Authors quote papers 51, 53 and 54 but it seems to me that the only relevant reference is 52 (Hergenrother et al., 2020). While ref. (53) doesn't seem very related to the choice of the phase function, the other papers (51) and (54) simply refer to the work by Hergenrother et al. (2020).

That is correct, we fixed it in the text.

4. p. 27 "is the particle magnitude at 1 AU", please add "from Sun and observer" (or something like that).

We added it to the manuscript.

REVIEWER 3

In the submitted paper, the authors examine the evolution of impact ejecta created by the DART spacecraft, using numerical simulations of ejected particles, generation of synthetic images and comparison of the results with the images captured by HST. The presented results are consistent with other works focusing on the DART impact. The process of formation of the ejecta features (an impact cone and a tail created by the SRP) has been described by Li et al. 2023, but to my knowledge it hasn't been studied extensively using numerical simulations. The authors further explain that the tail bifurcation observed in the HST images is a natural consequence of the system's dynamics and does not require secondary impacts, fragment disintegration or any other assumption. This work thus constitutes a valuable contribution and I recommend the paper for publication with the following remarks.

MAJOR COMMENTS

I have two questions for the authors which could be briefly addressed in the paper:

1. Using the synthetic images, would it be possible to further generate a synthetic light curve and compare it with the measured light curve of the Didymos system during the impact? As the measured light curve showed a temporary brightening by a factor 8.3 (Dotto et al. 2024), I'd be curious if a simulation that minimizes the photometric error also showed brightening by a similar factor.

Thanks for this comment. We believe this would be a very interesting study to be performed. However, we think this falls out of the scope of this paper, which is mainly focused on investigating the dynamical origin and morphology of observed post-impact features. We think a discussion about the brightening and its origin is an important matter, still debated in the scientific community, and therefore should be analysed carefully, possibly in a dedicated work.

2. The bifurcated tail naturally appears in the synthetic images, but the opening angle is notably larger than in the HST images - while the observed opening angle is around 4° , the synthetic images show an angle closer to 10° . What are some of the variables that can affect this? Could the opening angle be related to some of the simulation parameters? It would be helpful if the measured angle could further constrain some unknown parameters, such as the ejecta SFD, or even the mass of Dimorphos.

This is a very important remark. In our work, we investigate the basic dynamical mechanisms to form a bifurcation in the tail naturally (i.e., without any secondary event). We observe that, on a qualitative level, it is indeed possible to have such bifurcation as a consequence of natural dynamics alone. We also see some variability within the simulation campaign: the bifurcation occurs at different times and with different geometries (opening angles) for different initial condition sets.

As suggested by the reviewer, we analysed these data, searching for patterns or general trends. However, our dataset did not permit deriving any general consideration or quantitative indication on which parameters cause these differences and how. We definitely see some dependency on ejecta plume velocity-size distribution, but the data were too sparse and no obvious/unequivocal trend appeared from the analysis. We conclude that more data, with a wider and denser parameter grid, is required to quantify such effects.

We added some of these considerations to the manuscript.

MINOR ISSUES

1. Figure 3: The field of view indicated in the caption is not consistent with the length scale shown in the image. The length scale suggests the field of view is 20km.

Thanks for spotting this: that is correct, the FOV is +/- 10km, i.e., 20 km. This applies to supplementary videos as well. We fixed it in the caption and supplementary material.

2. Figure 5: The y-axis label should be “dSFD” to avoid ambiguity, especially since the term “SFD” is used in the Methods section for the integral of dSFD. There is further a typo in the b) plot (“SDF”).

Thanks for pointing this out, it was a typo in the figures y-axis label. The manuscript has been updated accordingly.

3. There are a few typos in the Methods section:

- Typo in Fig. 7 (Ejecta Dynamical Paramteres). The label of the Figure also says “77”.
- “... of 5 million ejecta particles in the, using ...”
- “... the cone-related and tail-relate features between ...”
- “Let $\pi N(r)$ the ejecta dSFD defined”
- “its integral its equal to 1.”

We fixed the typos in the manuscript.

4. Figure 8 uses the term “wing” in the legend, which is not used elsewhere in the paper. It would be better to unify the terminology.

The legend was revised using terminology of the manuscript: “cone-related” and “tail-related”.